# Proposal for simplified template cross-sections extension using $CP$ observables in $t\bar{t}H$

**Alberto Carnelli[1][†],**

**1** LAPP, Université Savoie Mont Blanc, CNRS/IN2P3, 74941 Annecy; France

† alberto.carnelli@cern.ch

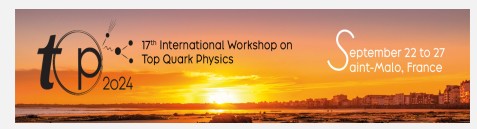

## Abstract

**The Large Hadron Collider (LHC) offers a unique opportunity to investigate $CP$ violation in the Yukawa coupling between the Higgs boson and the top quark by studying Higgs production in association with top quarks; this is of fundamental importance, seeing that the $CP$ properties of the Higgs boson are yet to measure with high precision. To address this, the focus of this work has been an extension of the simplified template cross-section (STXS) framework, devised to be sensitive to $CP$ effects. Our study focused on $CP$-sensitive observables across multiple Higgs decay channels, comparing their performances. The result indicates that the most efficient extension of the current binning used in the STXS framework, which currently uses the Higgs boson's transverse momentum $p_{T,H}$, requires adding one further split using $CP$-sensitive observables. Between these observables, one of the best is the Collins-Soper angle $|\cos\theta^*|$, a variable derived from momenta information of the top quarks. We have investigated the improvement brought by our two-dimensional STXS setup and compared it to the currently employed methodologies, finding an increase in performances at an integrated luminosity of 300 fb$^{-1}$. Moreover, our results highlight that this advantage seems to be present also at 3000 fb$^{-1}$.**

## 1 Introduction

Current observations indicate a baryon asymmetry in our Universe [1, 2], an asymmetry that can not be described by the Standard Model (SM) of particle physics. New sources of $CP$ violation not expected by the current SM theory are then needed and the search for corresponding $CP$-violating interactions is an essential target for searches beyond the SM (BSM) at the LHC. Recently, the $CP$ structure of the Higgs–fermion interactions has started to be probed [3, 4]. It has to be noted that BSM theories predict a larger amount of $CP$ violation in the Yukawa couplings with respect to other possible sources, like, for example, coming from interactions with massive vector bosons that are loop-suppressed. It is then of high importance to focus on identifying the $CP$ situation of Higgs–fermion

interactions, and of these, the top-Yukawa coupling has a special relevance being the highest. The Higgs Characterization Model [5] allows us to describe possible $\mathcal{CP}$ violation by varying the SM interaction as follows:

$$\mathcal{L}_{\text{top-Yuk}} = \frac{y_t^{\text{SM}} g_t}{\sqrt{2}} \bar{t} \left( \cos \alpha_t + i \gamma_5 \sin \alpha_t \right) t H \,. \tag{1}$$

In this parametrization:

- $y_t^{\text{SM}}$ is the SM top-Yukawa coupling,

- $g_t$ act as modifier of the strength of the top-Yukawa coupling,

- $\alpha_t$ is the $\mathcal{CP}$-mixing angle.

We retrieve the SM here if we consider $g_t = 1$ and $\alpha_t = 0$. This model has been used for direct searches in different Higgs decay channels. These searches have brought some initial limits but are tailored to the various channels and are not easy to combine. The simplified template cross-section (STXS) framework [6] has been explicitly established to mitigate issues like channel-specific setup. The objective of this work is, using $\mathcal{CP}$ observables, to enhance the sensitivity in $\mathcal{CP}$ of the STXS framework for the $t\bar{t}H$ production by adding an additional variable to the Higgs boson's transverse momentum $p_{T,H}$ currently employed.

## 2 Study setup

Using `MadGraph5_aMC@NLO` [7] we produced parton-level events for the $pp \to t\bar{t}H$ process at a center-of-mass energy of $\sqrt{s} = 13$ TeV. All samples were produced at leading order (LO) with one million events each, applying a scaling factor of 1.14 [8] as a next-to-leading-order (NLO) correction. We evaluate 11 $\mathcal{CP}$-discriminating observables across four reference frames:

- the laboratory frame (lab frame),

- the $t\bar{t}$ rest frame, where $\boldsymbol{p}_t + \boldsymbol{p}_{\bar{t}} = \boldsymbol{0}$ ($t\bar{t}$ frame),

- the $H$ rest frame, where $\boldsymbol{p}_H = \boldsymbol{0}$ ($H$ frame),

- the $t\bar{t}H$ rest frame, where $\boldsymbol{p}_t + \boldsymbol{p}_{\bar{t}} + \boldsymbol{p}_H = \boldsymbol{0}$ ($t\bar{t}H$ frame).

## 3 Performance evaluation

To extend the STXS framework, we focus on the three channels studied in current $t\bar{t}H$ analyses at the LHC that target different Higgs final states: $t\bar{t}H(\to \gamma\gamma)$, $t\bar{t}H(\to b\bar{b})$, and $t\bar{t}H(\text{multilep.})$, which is a way to group together $H(\to \tau\tau)$, $H(\to W^+W^-)$ and $H(\to ZZ)$ decays considering multiple leptons in the final state. We take the assumption to have access to measurements of the $t\bar{t}H$ distributions in each channel in order to measure the sensitivity of the observables discussed in Section 2 to $\mathcal{CP}$ violation. The effect of reconstruction effects (detector performance, etc) are taken into account by using scaling and smearing factors to the parton-level $t\bar{t}H$ events to produce realistic yields.

The sensitivity of an observable to a given BSM model is quantified through a significance $S$, as utilized by the ATLAS collaboration [9]. This significance reflects the power to distinguish the BSM hypothesis, parameterized by $g_t$ and $\alpha_t$, from the SM hypothesis, which implies $g_t = 1$ and $\alpha_t = 0$.

## 4  Results

The result presented here considers the whole dataset expected to be available at the end of LHC Run-3, corresponding to a luminosity of 300 fb$^{-1}$. The current experimental limits exclude $g_t = 1$ and $\alpha_t \gtrsim 43°$ at the 95% CL with 139 fb$^{-1}$ [4], as consequence a benchmark of $g_t = 1$ and $\alpha_t = 35°$ was chosen. We calculate the significance for all the studied observables and the associated two-dimensional observable combinations using 6 bins for every observable. The results indicate that using a combination of two observables gives higher performances. We favor combinations with $p_{T,H}$ due to the existing STXS binning and seeing that the results are similar to the optimal combination of other two observables. The best observables to combine with $p_{T,H}$ are: $\Delta\phi_{t\bar{t}}^{\text{lab}}$, $b_1^{\text{lab}}$, $b_2^{\text{lab}}$, $\Delta\eta_{t\bar{t}}^{t\bar{t}}$. After further bin optimization and background studies, we further reduced the candidate observables to $b_2^{\text{lab}}$, $\Delta\eta_{t\bar{t}}^{t\bar{t}}$, and $|\cos\theta^*|$. Based on this, we propose to extend the current STXS binning by adding one of the above observables; an example showing the improvement in exclusion limits adding $|\cos\theta^*|$ is presented in Fig 1. We also use a simplistic extrapolation to quantify the impact of this expanded framework on the exclusion limits at 3000 fb$^{-1}$, observing similar improvement. We performed a further study with Boosted Decision Trees (BDT), taking advantage of all the available $\mathcal{CP}$ observables, trying to obtain the maximal discrimination, showing only $\sim 10\%$ improvement in performances.

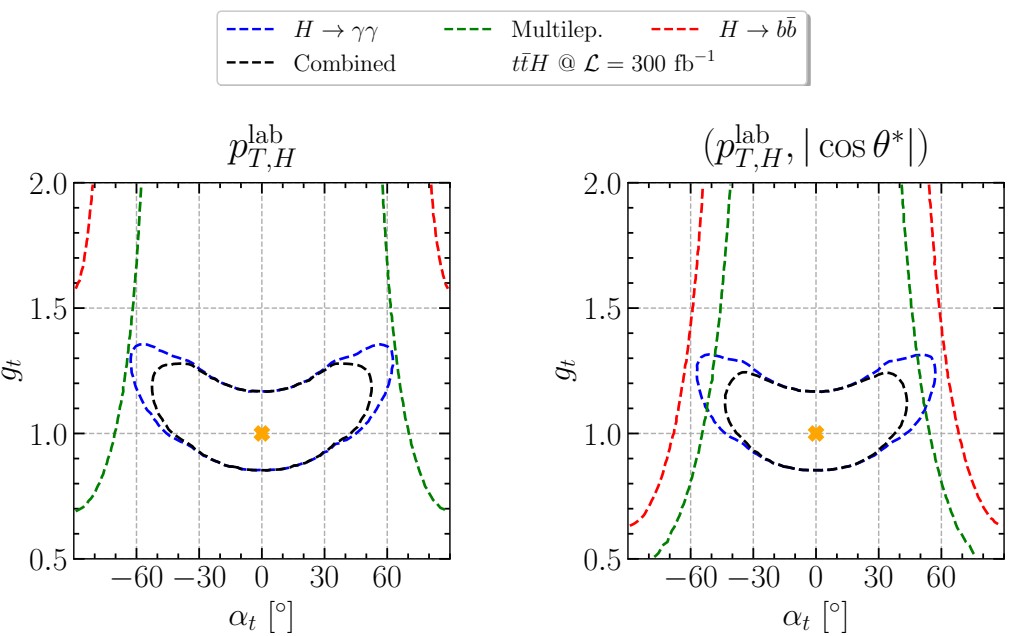

**Figure 1:** Limits obtained using our samples at the 95% confidence level in the plane $(\alpha_t, g_t)$ expected at the end of LHC Run-3 $\mathcal{L} = 300$ fb$^{-1}$ with the current $p_{T,H}$ STXS setup (left) and the improvement obtained with one of our 2-dimensional STXS extension: $(p_{T,H}, |\cos\theta^*|)$ (right).

## 5  Conclusion

To propose an extension of the current STXS v1.2 binning in $p_{T,H}$, we selected $\mathcal{CP}$ observables and studied their performances and combination; the results highlight that is ideal to extend the current STXS binning that uses the $p_{T,H}$ by combining with: $\Delta\eta_{t\bar{t}}$,

$|\cos\theta^*|$, and $b_2$. In the $(g_t, \alpha_t)$ plane, we have evaluated the exclusion limits using our proposed STXS extension versus the current STXS. For both 300 fb$^{-1}$ and 3000 fb$^{-1}$ of data, the proposed two-dimensional binning outperforms the current STXS binning and we also evaluate the result that can be obtained by employing a different methodology, a BDT, observing similar performances.

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
