# Peer review of "Proposal for simplified template cross-sections extension using $\cal{CP}$ observables in $t\overline{t}H$"

_SciPost Physics Proceedings_

## Round 1 · Referee Report · Anonymous (Referee 1) · 2025-4-14

Strengths

Topic of high relevance

Weaknesses

Too little detail on the study preformed; overly strong conclusions.

Report

This contribution investigates an extension of the STXS framework by binning in CP-sensitive quantities in the ttH channel. The aim is to get better access to possible CP violation in the top Yukawa coupling. The study is interesting and relevant for LHC Run3 as well as the high-luminosity phase of LHC operation. The presentation itself needs significant improvement, however. In particular more details need to be given on the binning used to obtain the results presented in Fig. 1 and the level of improvement needs to be better quantified. For detailed comments, see "requested changes" below. Clearly, points 6-9 are the most important ones; the page limit being 4 pages w/o references, there should be ample of space for addressing them.

Requested changes

1- Introduction: I suggest to rephrase the statement "that BSM theories predict a larger amount of CP violation in the Yukawa couplings with respect to other possible sources". While it is true that BSM theories can provide additional sources of CP violation beyond the SM, this is up to construction of the specific BSM theory, and not an imperative. So "can provide", "allow for" or similar seems more appropriate than "predict".

2- Bullet list below eq. (1): "g_t acts" (typo)

3- Last paragraph of section 1, "are tailored to the various channels and are not easy to combine." It is not directly clear the what is meant by by [search] channels. Searches in specific final states? Please make this specific. After all "channels" is used repeatedly thereafter.

4- Section 2: provide the MadGraph version number

5- Section 2, "applying a scaling factor of 1.14": A caveat that higher-order corrections might influence kinematic distributions seems in order here.

6- Section 3, "The effect of reconstruction effects (detector performance, etc) are taken into account by using scaling and smearing factors to the parton-level ttH events to produce realistic yields." is too vague. Provide details or appropriate references where these details are available (or directly the scaling and smearing functions in a code repository).

7- Section 4: define the "best observables" (what are b_1^lab etc.?) or provide a reference! And, why is cos\theta* not part of the list?

8- Provide the binning (bin sizes) used for obtaining the results in Fig. 1. For |cos\theta*| this is crucial. For p_T,H this is probably the STXS v1.2 binning, but it is better to give it explicitly. Also, how many events are there in each bin for the benchmark?

9- The improvement of limits on g_t and \alpha_t from adding |cos\theta*| is sizeable only in the b\bar b and multi-lepton channels. But, since the fit is dominated by the diphoton channel, the overall improvement is very mild, not to say marginal. This should be commented upon. Moreover, the combined result needs to be quantified in the text.

10- Conclusions, "the proposed two-dimensional binning outperforms the current STXS binning": as already requested in point 9, the "outperforming" needs to be quantified.

Recommendation

Ask for minor revision

---

## Round 2 · Referee Report · Anonymous (Referee 1) · 2025-6-10

Report

The points raised in my first report have been taken care of to satisfaction.

Recommendation

Publish (meets expectations and criteria for this Journal)

---

## Editorial Decision

accepted_in_target_journal